# Newly Synthesized Creatine Derivatives as Potential Neuroprotective and Antioxidant Agents on In Vitro Models of Parkinson’s Disease

**DOI:** 10.3390/life13010139

**Published:** 2023-01-04

**Authors:** Ivanka Kostadinova, Magdalena Kondeva-Burdina, Lyubomir Marinov, Lubomir L. Vezenkov, Rumyana Simeonova

**Affiliations:** 1Department of Pharmacology, Pharmacotherapy and Toxicology, Faculty of Pharmacy, Medical University of Sofia, 2 Dunav Str., 1000 Sofia, Bulgaria; 2Institut des Biomolécules Max Mousseron (IBMM), CNRS, Université de Montpellier, ENSCM, 226-234 Av. du Professeur Emile Jeanbrau, 34090 Montpellier, France

**Keywords:** creatine, antioxidant activity, neuroprotective effects, synaptosomes, mitochondria, microsomes

## Abstract

Oxidative stress is one of the key factors responsible for many diseases–neurodegenerative (Parkinson and Alzheimer) diseases, diabetes, atherosclerosis, etc. Creatine, a natural amino acid derivative, is capable of exerting mild, direct antioxidant activity in cultured mammalian cells acutely injured with an array of different reactive oxygen species (ROS) generating compounds. The aim of the study was in vitro (on isolated rat brain sub-cellular fractions–synaptosomes, mitochondria and microsomes) evaluation of newly synthetized creatine derivatives for possible antioxidant and neuroprotective activity. The synaptosomes and mitochondria were obtained by multiple centrifugations with Percoll, while microsomes–only by multiple centrifugations. Varying models of oxidative stress were used to study the possible antioxidant and neuroprotective effects of the respective compounds: on synaptosomes–6-hydroxydopamine; on mitochondria–tert-butyl hydroperoxide; and on microsomes–iron/ascorbate (non-enzyme-induced lipid peroxidation). Administered alone, creatine derivatives and creatine (at concentration 38 µM) revealed neurotoxic and pro-oxidant effects on isolated rat brain subcellular fractions (synaptosomes, mitochondria and microsomes). In models of 6-hydroxydopamine (on synaptosomes), tert-butyl hydroperoxide (on mitochondria) and iron/ascorbate (on microsomes)-induced oxidative stress, the derivatives showed neuroprotective and antioxidant effects. These effects may be due to the preservation of the reduced glutathione level, ROS scavenging and membranes’ stabilizers against free radicals. Thus, they play a role in the antioxidative defense system and have a promising potential as therapeutic neuroprotective agents for the treatment of neurodegenerative disorders, connected with oxidative stress, such as Parkinson’s disease.

## 1. Introduction

Parkinson’s disease (PD) is a progressive, and the second most common, neurodegenerative disease associated to aging [1,2,3]. Neurodegenerative changes in PD involve toxic mechanisms such as oxidative stress, mitochondrial dysfunction, apoptosis and inflammation [4]. Oxidative stress has been shown to cause extensive damage to lipids, proteins and deoxyribonucleic acid (DNA), resulting in cell death in relation to a variety of mechanisms, including activation of different apoptotic cell signaling molecules [5].

Nowadays, there are many in vitro and in vivo models which are used to mimic the pathological mechanisms in different neurodegenerative diseases. The most commonly used models for PD include induction with reserpine (in vivo) [6], 6-hydroxydopamine (6-OHDA)—in vitro and in vivo [7,8], or 1-methyl-4-phenyl-1,2,3,6-tetrahydropyridine (MPTP) (in vitro and in vivo) [9]. The human neuroblastoma cell line SH-SY5Y is also used for the PD in vitro model because it reproduces the dopaminergic phenotype typical for PD pathology [10]. Newer approaches for in vitro models of Alzheimer’s disease include the development of 3D models, the first of which was based on an immortalized human neural stem cell line (ReN) containing mutations in the amyloid precursor protein and Presenilin-1 genes [11]. Cell cultures based on oligodendrocytes or oligodendrocyte precursor cells from mice, rat and human origin have been developed to overcome the deficiencies of animal multiple sclerosis models. These cell lines are beneficial because they have the ability to differentiate and generate myelin in vitro in the absence of signals from axons [12].

6-OHDA is a highly oxidizable dopamine analog that can be captured through the dopamine transporter, which gives the neurotoxic agent specificity to affect catecholaminergic neurons, such as substantia nigra pars compacta dopaminergic neurons [13]. To date, three mechanisms have been proposed to explain the cytotoxic effect of 6-OHDA. These mechanisms include intra or extracellular auto-oxidation of 6-OHDA [14], formation of hydrogen peroxide (H_2_O_2_) [15] and direct inhibition of complex I of the mitochondrial respiratory chain [16]. Tert-Butyl hydroperoxide (t-BuOOH) is a classical lysosomal membrane permeabilization inducer that causes lysosomal damage via oxidative stress [17]. Microsomes can be used as a lipid membrane model to evaluate potential antioxidant activity. When ferrous sulfate is combined with ascorbic acid, a Fenton reaction takes place, which is associated with the generation of free hydroxyl radicals (•OH) [18]. These mechanisms can act independently or in combination to generate reactive oxygen species (ROS) [19]. The oxidative stress (OS) generated can be amplified by the increase in free calcium in the cytoplasm–a product of the excitotoxicity of glutamate, or by the loss of permeability of the mitochondrial membrane leading to cell death [20]. 

Endogenous antioxidants cannot completely prevent oxidative damage under physiological and pathological conditions. These conditions may lead to a disbalance in endogenous antioxidant systems and a subsequent surge in oxidative stress. In this case, the supplementation with antioxidant compounds such as creatine can have positive effects on the antioxidant system.

Creatine is formed by two essential amino acids, arginine and methionine, and one non-essential amino acid, glycine. Although the exact mechanism of action of creatine as an antioxidant is not fully understood, it has been shown to enhance the activity of antioxidant enzymes and the capability to eliminate ROS and reactive nitrogen species (RNS). The antioxidant effects of creatine may be related to different mechanisms such as increased or normalized cell energy status and the maintaining of mitochondrial integrity [21].

According to the data about antioxidant activity of creatine and the critical role of the oxidative stress in the pathogenesis of PD, it is interesting to evaluate the possible antioxidant (malondialdehyde (MDA) production) and neuroprotective effects (synaptosomal viability and reduced glutathione (GSH) content) of newly synthetized creatine derivatives on a variety of models of neurotoxicity (6-hydroxydopamine- and tert-butyl hydroperoxide-induced oxidative stress as well as non-enzyme-induced lipid peroxidation) in different subcellular fractions (isolated rat brain synaptosomes, mitochondria and microsomes).

## 2. Materials and Methods

### 2.1. Animals

The experiments were performed on nine, male white Wistar rats (300–350 g body weight), age 1.5–2 years, as per preceding experimental protocols. The animals were obtained from the National Breeding Center at the Bulgarian Academy of Sciences, Slivnitsa, Bulgaria, and bred under standard conditions in plexiglass cages with free access to water and food and 12 h/12 h light/dark cycle at a temperature of 20–25 °C. The experiments were carried out in accordance with Ordinance No. 20/1 November 2012 on the minimal requirements for the protection and welfare of experimental animals, and the European Directive 2010/63/EU on the protection of animals used for scientific purposes. The experiments were approved by The Bulgarian Food Safety Agency with Permit number 187, which is valid until 2023.

### 2.2. Materials

Chemicals that were used in our investigation were delivered by Sigma Aldrich, Germany: 3-(4,5-dimethylthiazol-2-yl)-2,5-diphenyltetrazolium bromide (MTT), Percoll, 4-(2-hydroxyethyl)-1-piperazineethanesulfonic acid (HEPES), sucrose, 2-thiobarbituric acid, Tris hydrochloride, dithiothreitol, phenylmethylsulfonyl fluoride and EDTA. NaCl, KCl, CaCl_2_·2H_2_O, MgCl_2_·2H_2_O, NaHPO_4_, D-glucose, trichloroacetic acid, 5,5′-dithiobis-(2-nitrobenzoic acid) (DTNB) and sulfuric acid were delivered by Merck, Germany.

The tested compounds–Creatine lysinate (Cr lysinate), Creatine ornitate (Cr ornitate), Creatine leucinate (Cr leucinate), Creatine isoleucinate (Cr isoleucinate) and Creatine valinate (Cr valinate) were synthesized by Prof. Lyubomir T. Vezenkov from the University of Chemical Technology and Metallurgy in Sofia, Bulgaria and analyzed by Assoc. Prof. Lubomir L. Vezenkov from IBMM, Montpellier, France (Patent for invention N 66511, Creatine salts, L. T. Vezenkov, P. T. Angelov, L. L. Vezenkov, published in Bulletin 11 on 30 November 2015).

The creatine salts according to the invention are with the amino acids of general formula:
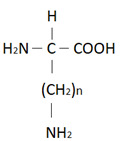

where n is a number from 1 to 4. The compounds were dissolved in dimethyl sulfoxide. The concentration used (38 μM) was calculated based on data from previous in vivo experiments. The subcellular fractions were incubated with creatine derivatives for 1 h. In models of toxicity, the incubation period was 1.5 h.

### 2.3. Isolation of Rat Brain Synaptosomes and Mitochondria

Synaptosomes and brain mitochondria were isolated by multiple differential centrifugations by the methods of Taupin et al. [22] and Sims & Anderson [23]. During the experiment, two buffers–A (HEPES 5 mM and Sucrose 0.32 M) and B (NaCl 290 mM, MgCl_2_·2H_2_O 0.95 mM, KCl 10 mM, CaCl_2_·2H_2_O 2.4 mM, NaH_2_PO_4_ 2.1 mM, HEPES 44 mM, D-Glucose 13 mM) were prepared. Buffer A was used to prepare the brain homogenate. First, the homogenate was centrifuged twice at 1000× *g* for 10 min at 4 °C, after which the supernatants of the two centrifuges were combined and centrifuged three times at 10,000× *g* for 20 min at 4 °C. Isolation of synaptosomes and mitochondria was accomplished through the use of Percoll. First, 90% Percoll stock solution was prepared and after that, 16% and 10% solutions were prepared. An amount of 4 mL of 16% and 10% Percoll were carefully applied in layers. At the end, 4 mL of 90% Percoll (7.5% Percoll) was added to the precipitate from the last centrifugation. The tubes were centrifuged at 15,000× *g* for 20 min at 4 °C. After centrifugation, three layers were formed: the lower, which contains mitochondria; medium (between 16% and 10% Percoll), containing synaptosomes; and at the top, lipids. Using a Pasteur pipette, we removed the middle and bottom layers. Each was centrifuged at 10,000× *g* for 20 min at 4 °C with buffer B. The resulting synaptosomes and mitochondria were diluted with buffer B to a protein content of 0.1 mg/mL.

Synaptosomes were incubated with 150 µM 6-hydroxydopamine [24] and mitochondria–with 75 µM tert-butyl hydroperoxide [25].

### 2.4. Synaptosomal Viability

After incubation with test substances (Cr lysinate, Cr ornitate, Cr leucinate, Cr isoleucinate, Cr valinate, Creatine) and 6-OHDA, synaptosomes were centrifuged three times at 15,000× *g* for 1 min. Synaptosomal viability was determined by an MTT test by the method of Mungarro-Menchaca et al. [26].

### 2.5. Determination of GSH Levels in Synaptosomes and Mitochondria

After incubation with the substances (Cr lysinate, Cr ornitate, Cr leucinate, Cr isoleucinate, Cr valinate, Creatine), the synaptosomes were centrifuged at 400× *g* for 3 min. The precipitate was treated with 5% trichloroacetic acid, vortexed and left on ice for 10 min, then centrifuged at 8000× *g* for 10 min. The supernatant was frozen at −20 °C. Immediately prior to determination, each sample was neutralized with 5 M NaOH. GSH levels were determined by Elmman’s reagent (DTNB) spectrophotometrically at 412 nm by methods of Robyt et al. [27] and Shirani et al. [28].

For determination of GSH in synaptosomes and mitochondria, creatine was used as positive control. In isolated rat brain synaptosomes, pure 6-OHDA was used for negative control and in isolated rat brain mitochondria–pure tert-butyl hydroperoxide was used.

### 2.6. Isolation of Brain Microsomes

The brain was homogenized in 9 volumes of 0.1 M Tris buffer containing 0.1 mM Dithiothreitol, 0.1 mM Phenylmethylsulfonyl fluoride, 0.2 mM EDTA, 1.15% KCl and 20% (*v*/*v*) glycerol (pH 7.4). The resulting homogenate was centrifuged twice at 17,000× *g* for 30 min. The supernatants from the two centrifuges were combined and centrifuged twice at 100,000× *g* for 1 h. The pellet was frozen in 0.1 M Tris buffer at −20 °C [29].

### 2.7. Determination of MDA in Brain Mitochondria and Microsomes

The determination of MDA is spectrophotometrically at a wavelength of 535 nm by methods of Shirani et al. [28] and Mansuy et al. [18]. A molar extinction coefficient of 1.56 × 105 M^−1^ cm^−1^ is used for the calculation [30].

### 2.8. Statistical Analysis

The results of the experiments were statistically processed using the statistical software GraphPad Prism 8, using the non-parametric Mann–Whitney method at degrees of significance *p* < 0.05, *p* < 0.01 and *p* < 0.001.

## 3. Results

### 3.1. Effects of Creatine Derivatives (at Concentration 38 µM) on Isolated Rat Brain Synaptosomes

Creatine derivatives and creatine show neurotoxic effects on synaptosomes. They reduced synaptosomal viability statistically significantly by 35% (*p* < 0.05) and 30% (*p* < 0.05), and GSH levels by 25% (*p* < 0.05) and 20% (*p* < 0.05), respectively, compared to controls (untreated synaptosomes) (Figure 1 and Figure 2).

6-hydroxydopamine (6-OHDA) alone showed a statistically significant neurotoxic effect by reducing synaptosomal viability and GSH levels by 55% (*p* < 0.001) and 50% (*p* < 0.001), respectively, compared to the control group (untreated synaptosomes) (Figure 1 and Figure 2).

In a model of 6-OHDA-induced oxidative stress, creatine derivatives and creatine showed a statistically significant neuroprotective effect. The substances maintained synaptosomal viability statistically significantly by 71% (*p* < 0.01) and 73% (*p* < 0.01), and GSH levels by 70% (*p* < 0.05) and 80% (*p* < 0.05), respectively, compared to 6-OHDA (Figure 1 and Figure 2).

### 3.2. Effects of Creatine Derivatives (at Concentration 38 µM) on Isolated Brain Mitochondria

Individually, creatine derivatives and creatine show neurotoxic effect on mitochondria. The compounds reduced GSH levels statistically significantly by 25% (*p* < 0.05) and 20% (*p* < 0.05), and increased MDA production by 32% (*p* < 0.05) and 44% (*p* < 0.05), respectively, compared to the control group (untreated mitochondria) (Figure 3 and Figure 4). Administered alone, t-BuOOH showed a statistically significant neurotoxic effect by reducing GSH levels by 50% (*p* < 0.001) and increasing MDA production by 152% (*p* < 0.001) compared to the control group (untreated mitochondria) (Figure 3 and Figure 4). In a model of t-BuOOH-induced oxidative stress, creatine derivatives and creatine showed a statistically significant neuroprotective effect (*p* < 0.01). The substances maintained the GSH level statistically significantly by 70% (*p* < 0.01) and 80% (*p* < 0.01) and reduced MDA production by 39% (*p* < 0.01) and 35% (*p* < 0.01), respectively, compared to t-BuOOH (Figure 3 and Figure 4).

### 3.3. Effects of Creatine Derivatives (at Concentration 38 µM) on Isolated Brain Microsomes

On microsomes, administered alone, creatine derivatives and creatine show a significant pro-oxidant effect. They increased MDA production statistically significantly by 35% (*p* < 0.05) and 43% (*p* < 0.05), respectively, compared to the control group (untreated microsomes) (Figure 5). Under conditions of non-enzymatic lipid peroxidation, a statistically significant increase in the production of MDA by 130% (*p* < 0.001) was observed compared to the control (untreated microsomes). In this model, creatine derivatives and creatine exhibit a statistically significant antioxidant effect. The substances reduced MDA production statistically significantly by 45% (*p* < 0.001) and 42% (*p* < 0.001), respectively, compared to the toxic agent (Figure 5).

## 4. Discussion

Creatine has been widely used as a dietary supplement to improve physical performance in humans and also to ameliorate several neurodegenerative diseases associated with glutamatergic excitotoxicity and oxidative stress [30]. Creatine supplementation shows in vitro neuroprotective effects against different neurotoxins [31] and it can slow down the progression of some neurodegenerative conditions, as revealed previously [32]. There are documented clinical trials which insinuate that creatine supplementation could have beneficial effects in diseases such as Huntington’s, amyotrophic lateral sclerosis, depression or PD [33,34].

In our models, creatine derivatives and creatine show statistically significant neurotoxic effects. There is data implying that creatine supplementation promotes free radical generation and induction of oxidative stress in healthy athletes. Percário et al. [35] examined the effect of creatine monohydrate supplementation and resistance training on muscle strength and oxidative stress profiles in healthy athletes. They concluded that creatine supplementation associated to a specific resistance program promotes a significant increase in muscular strength. However, the significant increase in uric acid and the decrease in the total antioxidant status in creatine monohydrate supplemented subjects hints that creatine supplementation promotes free radical generation. This indicates that creatine supplementation, despite promoting improvement of muscle strength, may also have negative effects owing to the induction of oxidative stress and a decrease in total antioxidant status.

Three main mechanisms are involved in 6-OHDA-induced dopaminergic cell death: ROS generation by intra or extracellular auto-oxidation; H_2_O_2_ formation induced by monoamine oxidase activity; and direct inhibition of the mitochondrial respiratory chain, leading to oxidative stress, a decrease in cellular adenosine triphosphate (ATP) availability [36], and cell death [14]. The results from our study revealed that 6-hydroxydopamine alone showed a statistically significant neurotoxic effect by reducing synaptosomal viability and GSH levels by 55% and 50%, respectively, compared to the control group (untreated synaptosomes).

In a model of 6-OHDA-induced oxidative stress in our study, creatine derivatives and creatine showed a statistically significant neuroprotective effect. The substances maintained synaptosomal viability statistically significantly by 71% and 73%, and GSH levels by 70% and 80%, respectively, compared to the control group (untreated synaptosomes).

Creatine has been reported to exert beneficial effects in several neurodegenerative diseases in which glutamatergic excitotoxicity and oxidative stress have a crucial role. Cunha et al. [30] conducted an inquiry into the protective effects of creatine, as compared to the N-Methyl-D-Aspartate (NMDA) receptor antagonist dizocilpine (MK-801), against glutamate or H_2_O_2_-induced injury in human neuroblastoma SH-SY5Y cells. They revealed that creatine (1–10 mM) or MK-801 (0.1–10 μM) reduced glutamate- and H_2_O_2_-induced toxicity. The protective effect of creatine against glutamate-induced toxicity involves its antioxidant effect, since creatine, similar to MK-801, prevented the increase on dichlorofluorescein fluorescence induced by glutamate or H_2_O_2_.

In another experiment, the repeated administration of creatine (300 mg/kg, p.o.) in mice prevented the decreases on cellular viability and mitochondrial membrane potential of hippocampal slices incubated with glutamate (10 mM). The authors stated that creatine decreased the amount of nitrite formed in the reaction of oxygen with NO produced from the sodium nitroprusside solution. It is possible that the protective effect of creatine against glutamate or H_2_O_2_-induced toxicity might be due to its scavenger activity. These results suggested that creatine may be useful as an adjuvant therapy for neurodegenerative diseases [30].

Several mechanisms of action of t-BuOOH are described: formation of reactive free radicals [25], decrease the activity of glutathione reductase and increase the activity of glutathione peroxidase, which leads to a decrease in the amount of GSH. Other mechanisms include an increase in the intracellular concentration of calcium, which promotes cell apoptosis and oxidation of sulfhydryl (-SH) groups of mitochondrial enzymes and the inhibition of cellular respiration [37].

Malondialdehyde (MDA) as an end product of lipid oxidation is considered to be a reliable indicator of ROS formation and oxidative stress [17]. The MDA levels were measured to look into the effect of creatine and creatine derivatives on t-BuOOH-induced lipid peroxidation. A significant elevation (by 152%) of the MDA contents was observed in isolated mitochondria with 75 μM of t-BuOOH, compared with the control group, whereas cotreatment with 38 µM of creatine and creatine derivatives exhibited a significant decrease (by 39% and 35%, respectively) in lipid peroxidation. Furthermore, creatine and creatine derivatives at a concentration of 38 µM maintained the GSH level statistically significant by 70% and 80%, respectively, whereas t-BuOOH, administered alone, showed a statistically significant neurotoxic effect by reducing GSH levels by 50%. The results showed that co-treatment of isolated mitochondria with creatine and creatine derivatives inhibited t-BuOOH-induced MDA production, alleviated lipid peroxidation of the cell membrane and reduced cell damage.

Guidi et al. [38] reported the protective effect of creatine on mitochondrial DNA. In their investigation, they used an acute oxidative stress model and found that mitochondrial DNA is protected by creatine. Their results from cellular experiments suggest that creatine supplementation may play an important role in mitochondrial genome stability in that it could normalize mitochondrial mutagenesis. This can have functional consequences such as the decrease of oxygen consumption, mitochondrial membrane potential and ATP content, and finally cell survival. This can be traced to the biochemical features of creatine; the authors concluded that it could be a promising antigenotoxic agent for the treatment of the oxidative stress-related human diseases and in neuropathologies, such as Huntington disease, Parkinson disease and amyotrophic lateral sclerosis, as well as for the delaying of aging.

Another creatine mechanism is related to the regulation of calcium homeostasis. Impaired sarcoplasmic reticulum due to muscle damage may increase calcium concentrations in the cytosol, causing secondary muscle damage [39]. Creatine participates in maintaining the sarcoplasmic reticulum calcium pump function by phosphorylating adenosine diphosphate to adenosine triphosphate, which leads to a decrease in cytosolic calcium levels [40,41].

Valerio et al. [42] explored the effects of a balanced amino acid mixture with a high content of branched-chain and other essential amino acids (BCAA-enriched mixture, BCAAem; % composition: leucine 31.3, lysine 16.2, isoleucine 15.6, valine 15.6, threonine 8.8, cysteine 3.8, histidine 3.8, phenylalanine 2.5, methionine 1.3, tyrosine 0.7, tryptophan 0.5) which have been found to improve age-related disorders in animals and humans. The authors demonstrated that BCAAem oral supplementation increases the average, but not maximal, lifespan of male mice and enhanced mitochondrial biogenesis and function in cardiac and skeletal muscles. The key role of BCAAem on mitochondrial biogenesis, cell energy metabolism and ROS scavenging systems may explain most of the beneficial actions of this supplementation.

Microsomes are heterogeneous vesicle-like fragments with a diameter of 20–200 nm, which are formed in vitro from parts of the endoplasmic reticulum during fragmentation of eukaryotic cells. Microsomes are not observed in healthy, living cells. Microsomes can be concentrated and separated from other cellular organelles by repeated centrifugation. They have a reddish-brown color due to the presence of heme. They represent an in vitro model for analyzing the metabolic activity of Cytochrome P450 (CYP) enzymes. Microsomes are widely used as experimental systems to study metabolic stability and metabolic profile during the discovery and development of new drugs. Microsomal liver fraction combined with the ability to automate the incubation process are included in high-tech applications. The ability of microsomes to store most of the enzymes involved in phases I and II of biotransformation allows for the creation of a bank for the study of genetic polymorphism. They can be used as a model of lipid membrane in experiments for investigating the possible antioxidant activity of different biologically active substances from natural or synthetic origins [24].

Creatine derivatives and creatine show pro-oxidant effects on the microsomes. The compounds increased MDA production statistically significantly by 35% and 43%, respectively, compared to the control group (untreated microsomes). Under conditions of non-enzyme-induced lipid peroxidation, a statistically significant increase in the production of MDA by 130% was observed compared to the control (untreated microsomes). In the model with the pro-oxidant ferrous ascorbate, creatine derivatives and creatine exhibit a statistically significant antioxidant effect. They reduced MDA production statistically significantly by 45% and 42%, respectively, compared to the toxic agent. Creatine has been considered a cytoplasmic antioxidant of direct action that would mainly promote the scavenging of ROS [43]. In their study, Sestili et al. [44] proved that creatine exerts direct antioxidant activity in cultured mammalian cells exposed to various oxidizing agents. The antioxidant activity of creatine and its derivatives on isolated rat brain microsomes might be due to its direct action as an ROS scavenger.

Although we used in vitro models of neurotoxicity with different mechanisms to prove the antioxidant potential of creatine as a possible mechanism for its role as neuroprotector, there are some limitations to our study. The first is to check the morphology of synaptosomes and the second–to investigate the effects of creatine on the in vivo model of Parkinson’s disease, induced by rotenone in mice.

## 5. Conclusions

Administered alone, creatine derivatives and creatine (at a concentration of 38 µM) show a neurotoxic and pro-oxidant effect on isolated rat brain subcellular fractions (synaptosomes, mitochondria and microsomes). In alternate neurotoxicity models, provoked by oxidative stress, induced by 6-OHDA, t-BuOOH and ferro ascorbate, the compounds show a neuroprotective and antioxidant effect on isolated synaptosomes, mitochondria and microsomes. These neuroprotective and antioxidant effects of creatine and creatine derivatives might be due to the preservation of GSH levels on one hand and direct ROS scavenging on the other. Thus, they play a role in the antioxidative defense system and have a promising potential as therapeutic neuroprotective agents for the treatment of neurodegenerative disorders, connected with oxidative stress, such as PD.

## Figures and Tables

**Figure 1 life-13-00139-f001:**
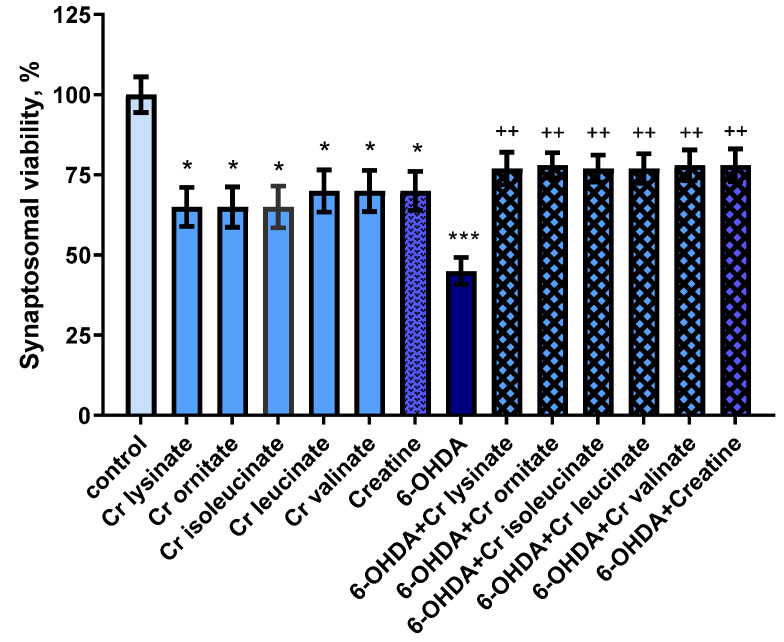
Effects of creatine derivatives (concentration 38 µM) administered alone and in 6-OHDA-induced oxidative stress model, on synaptosomal viability. The non-parametric Mann–Whitney method showed: * *p* < 0.05; *** *p* < 0.001 vs. control (untreated synaptosomes); ++ *p* < 0.01 vs. 6-OHDA.

**Figure 2 life-13-00139-f002:**
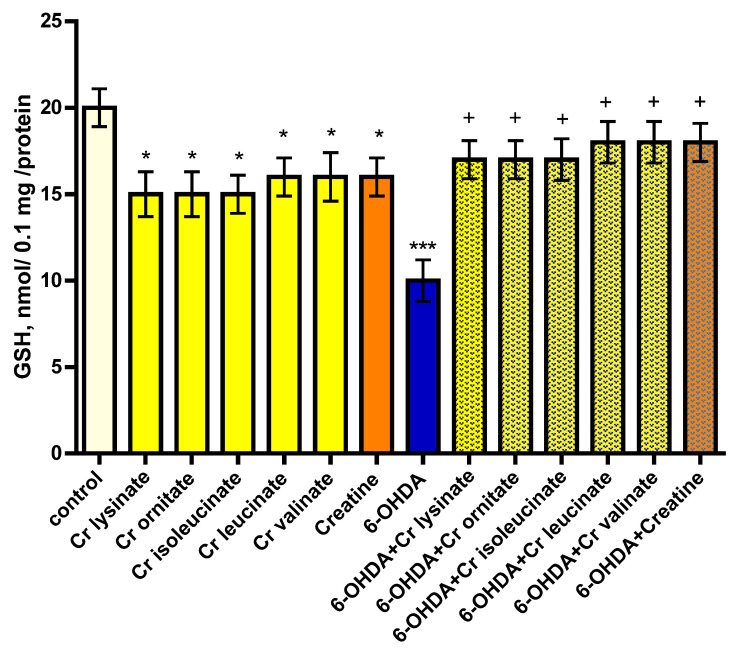
Effects of creatine derivatives (concentration 38 µM) administered alone and in 6-OHDA-induced oxidative stress model, on GSH levels in synaptosomes. The non-parametric Mann–Whitney method showed: * *p* < 0.05; *** *p* < 0.001 vs. control (untreated synaptosomes); + *p* < 0.05 vs. 6-OHDA.

**Figure 3 life-13-00139-f003:**
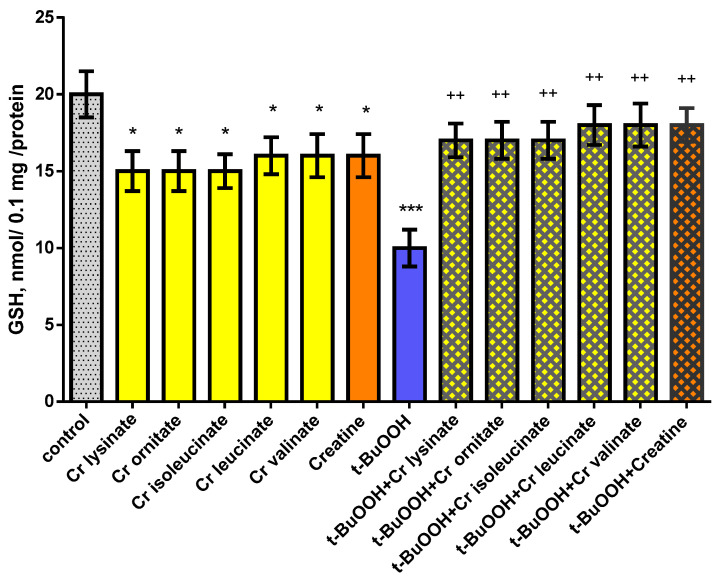
Effects of creatine derivatives (concentration 38 µM) administered alone and in t-BuOOH-induced oxidative stress model, on GSH levels in isolated mitochondria. The non-parametric Mann–Whitney method showed: * *p* < 0.05; *** *p* < 0.001 vs. control group (untreated mitochondria); ++ *p* < 0.01 vs. t-BuOOH.

**Figure 4 life-13-00139-f004:**
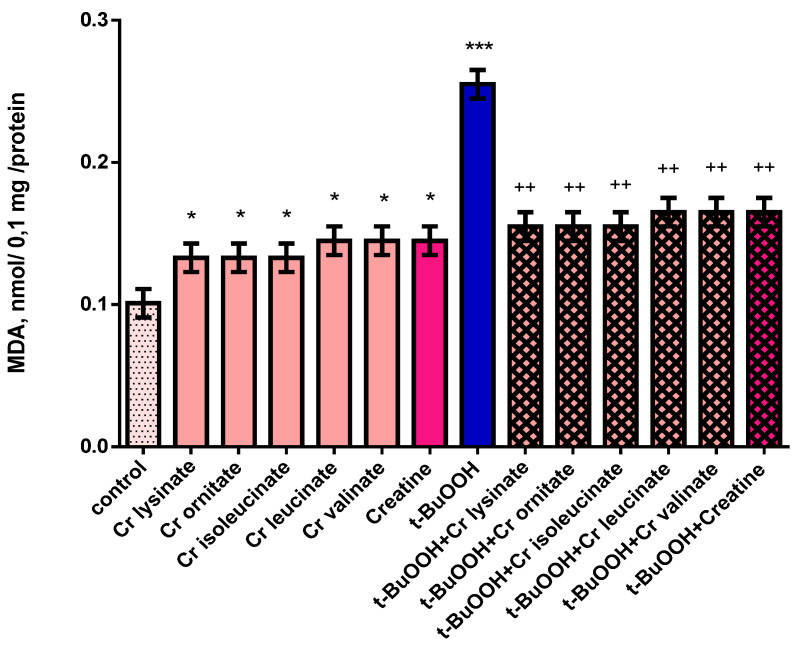
Effects of creatine derivatives (concentration 38 µM) administered alone and in t-BuOOH-induced oxidative stress model, on MDA production in isolated brain mitochondria. The non-parametric Mann–Whitney method showed: * *p* < 0.05; *** *p* < 0.001 vs. control group (untreated mitochondria); ++ *p* < 0.01 vs. t-BuOOH.

**Figure 5 life-13-00139-f005:**
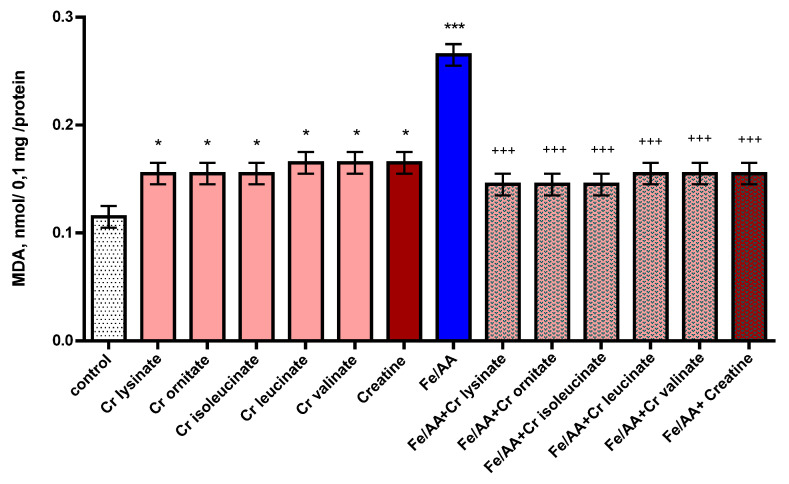
Effects of creatine derivatives (concentration 38 µM) administered alone and in non-enzyme-induced lipid peroxidation model, on MDA production in isolated brain microsomes. The non-parametric Mann–Whitney method showed: * *p* < 0.05; *** *p* < 0.001 vs. control group (untreated microsomes); +++ *p* < 0.001 vs. toxic agent (Fe/AA).

## Data Availability

Not applicable.

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
