# Peer review of "Newly Synthesized Creatine Derivatives as Potential Neuroprotective and Antioxidant Agents on In Vitro Models of Parkinson’s Disease"

_life, 2023, doi:10.3390/life13010139_

Round 1

Reviewer 1 Report (Previous Reviewer 1)

Comments for life-2089599

The manuscript aimed to in vitro evaluation of newly synthetized creatine derivatives for possible antioxidant and neuroprotective activity on isolated rat brain sub-cellular fractions – synaptosomes, mitochondria and microsomes. In the revised manuscript, my questions in experimental design has been responded by authors. Thus I suggest that the manuscript be considered for publication.

Author Response

Thank you very much for your evaluation!

Best regards,

Simeonova

Reviewer 2 Report (Previous Reviewer 2)

Title:

1.     Please fix the grammatical errors in the title.

Abstract:

1.     Line 18-19, The aim of the study was in vitro evaluation...... Please explain the word “in vitro” here.

Introduction

1.     Please correct the reference style as per the journal requirements. Line 52 and 56.

2.     Authors need to provide a bird’s eye view of their research in the last part of the introduction.

Materials and methods:

1.     The weight of the experimental animals is missing. Why did the authors use advanced-age rats (1.5-2 years old)?

2.     The chemical formulas need to be corrected. Line numbers 99, 108.

3.     What type of tested compounds are the authors referring to? Lines 102-103.

4.     Please specify the substances. Kindly refer to lines 126, and 130.

5.     It is strange to see a reference in a heading (line 139). It can be provided in the detail of the title.

Discussion

1.     The argument needs to be supported by a proper reference. (lines 226-227)

There are a lot of grammatical mistakes and the overall sentence structure is poor. Authors need to get the services of a professional English editor.

Author Response

We would like to thank the reviewer for reviewing our manuscript and providing constructive comments. The answers to the comments is detailed below.

We have added one more co-author - Lyubomir L. Vezenkov²

All corrections are colored in blue.

Questions/ Comments

Answer

Please fix the grammatical errors in the title.

The grammatical error is fixed

All corrections are colored in blue

Line 18-19, The aim of the study was in vitro evaluation...... Please explain the word “in vitro” here.

in vitro (on isolated rat brain sub-cellular fractions – synaptosomes, mitochondria and microsomes)

Please correct the reference style as per the journal requirements. Line 52 and 56.

The reference style was corrected

  Authors need to provide a bird’s eye view of their research in the last part of the introduction.

We provided a bird’s eye view and provided an additional information about the research:

According the data about antioxidant activity of creatine and the critical role of the oxidative stress in the pathogenesis of PD, it’s interesting to evaluate the possible antioxidant (malondialdehyde (MDA) production) and neuroprotective effects (synaptosomal viability and reduced glutathione (GSH) content) of newly synthetized creatine derivatives on different models of neurotoxicity (6-hydroxydopamine- and tert-butyl hydroperoxide-induced oxidative stress as well as non-enzyme-induced lipid peroxidation) in different subcellular fractions (isolated rat brain synaptosomes, mitochondria and microsomes).

The weight of the experimental animals is missing. Why did the authors use advanced-age rats (1.5-2 years old)?

According to the experimental protocol the use of advanced-age rats is needed.

The chemical formulas need to be corrected. Line numbers 99, 108.

The chemical formulas were corrected

What type of tested compounds are the authors referring to? Lines 102-103.

The tested compounds are: Creatine lysinate, Creatine ornitate, Creatine leucinate, Creatine isoleucinate, Creatine valinate, Creatine

Please specify the substances. Kindly refer to lines 126, and 130.

Creatine lysinate, Creatine ornitate, Creatine leucinate, Creatine isoleucinate, Creatine valinate, Creatine

It is strange to see a reference in a heading (line 139). It can be provided in the detail of the title.

A reference is provided after the text

The argument needs to be supported by a proper reference. (lines 226-227)

A proper reference was added

There are a lot of grammatical mistakes and the overall sentence structure is poor. Authors need to get the services of a professional English editor.

The current text has been checked for grammatical errors by a native English speaker.

Best regards,

Simeonova

Reviewer 3 Report (New Reviewer)

The manuscript by Kostadinova et al. entitled "Newly synthetized creatine derivatives as potential neuroprotective and antioxidant agents on in vitro models of Parkinson’s disease" presents studies which evaluate neuroprotective and antioxidant potential of creatine derivatives. The results are novel and noteworthy. However, in my opinion this manuscript needs to be rewritten and clarified. Please see my comments below.

1. The Authors are inconsequent in the use of abbreviations. Some abbreviation are not explained at all. At first use the abbreviation should be explained and then used consistently throughout the manuscript.

2. Materials section: I think it should be renamed, e.g. Substances, and should be rewritten. Authors should provide what creatine derivatives were used and how their solutions were prepared. It would be worth considering adding a research schedule.

3. Materials: subscript should be used when giving chemical formulas of compounds.

4. Authors should not use the following phrases in the Materials and Methods section: “Some of the chemicals…”, “Other chemicals like…”.

5. Why the Authors chose the concentration of 38 μM for their research?

6. How long were the mitochondria, synaptosomes and microsomes incubated in solutions of creatine and creatine derivates?

7. Subsection 2.6. is “Isolation of brain microsomes [26]” should be “Isolation of brain microsomes”. Literature should be cited in the text, not in the subsection title.

8. Subsection 2.7. is “Determination of malondialdehyde (MDA) in brain mitochondria and microsomes [34]” should be “Determination of malondialdehyde (MDA) in brain mitochondria and microsomes”. Literature should be cited in the text, not in the subsection title.

9. Why Authors used creatine as a positive control?

10. Creatine should not be mentioned in the titles of the subsections and the titles of the figures, as the assessment of its activity was not the purpose of presented research. Creatine was only a control group in this study.

11. Results section: This section is not informative enough. More statistical parameters should be included. The Authors give only the % level of the tested parameters.

12. Terms such as “weak statistically significant”, “low statistically significant” or “little statistically significant” should not be used to describe the obtained results of experiments. Authors should use p-values in the description of the outcomes.

13. Figures: It is not necessary to write the dose of the tested substance in each column, especially when it is the same everywhere. This information can be included in the caption to the figure.

14. The cited literature is quite old. Is there no more recent literature data?

15. L. 29 “good” isn’t appropriate word if we are talking about the neuroprotective and antioxidant effects

16. L. 46 “treatment” isn’t appropriate word if we are writing about model, maybe it should be …include models induced by …

17. L. 46 Please provide whether indicated models are animal or in vitro PD models?

18. L. 49,52,56 There's something wrong with the citation

Author Response

Journal name: Life

MS ID: life-2089599

Title: Evaluation of in vitro neuroprotective and antioxidant activity of newly synthetized creatine derivatives on different sub-cellular fractions (rat brain synaptosomes, mitochondria and microsomes)

Authors: Ivanka Kostadinova1, Magdalena Kondeva-Burdina1, Lyubomir Marinov1, Lyubomir L. Vezenkov², Rumyana Simeonova1*

We would like to thank the reviewer for reviewing our manuscript and providing constructive comments. The answers to the comments is detailed below.

We have added one more co-author - Lyubomir L. Vezenkov²

All corrections are colored in blue

Questions/ Comments

Answer

1. The Authors are inconsequent in the use of abbreviations. Some abbreviation are not explained at all. At first use the abbreviation should be explained and then used consistently throughout the manuscript.

Now all abbreviations are explained when they first appear in the text

All corrections are colored in blue

2. Materials section: I think it should be renamed, e.g. Substances, and should be rewritten. Authors should provide what creatine derivatives were used and how their solutions were prepared. It would be worth considering adding a research schedule.

The tested compounds are: Creatine lysinate, Creatine ornitate, Creatine leucinate, Creatine isoleucinate, Creatine valinate, Creatine

The compounds were dissolved in dimethyl sulfoxide (DMSO)

3. Materials: subscript should be used when giving chemical formulas of compounds.

The chemical formulas were corrected

4. Authors should not use the following phrases in the Materials and Methods section: “Some of the chemicals…”, “Other chemicals like…”.

The phrases were corrected

5. Why the Authors chose the concentration of 38 μM for their research?

The concentration used (38 μM) was calculated based on data from previous in vivo experiments.

6. How long were the mitochondria, synaptosomes and microsomes incubated in solutions of creatine and creatine derivates?

When administered alone, the subcellular fractions were incubated with creatine deriva-tives for 1 hour. In models of toxicity, the incubation period is 1.5 hour.

7. Subsection 2.6. is “Isolation of brain microsomes [26]” should be “Isolation of brain microsomes”. Literature should be cited in the text, not in the subsection title.

A reference is provided after the text

8. Subsection 2.7. is “Determination of malondialdehyde (MDA) in brain mitochondria and microsomes [34]” should be “Determination of malondialdehyde (MDA) in brain mitochondria and microsomes”. Literature should be cited in the text, not in the subsection title.

A reference is cited after the text

9. Why Authors used creatine as a positive control?

We used creatine as a positive control because newly synthesized compounds are derivatives of creatine

10. Creatine should not be mentioned in the titles of the subsections and the titles of the figures, as the assessment of its activity was not the purpose of presented research. Creatine was only a control group in this study.

In the titles of the subsections and the titles of the figures are mentioned only the creatine derivatives

11. Results section: This section is not informative enough. More statistical parameters should be included. The Authors give only the % level of the tested parameters.

In the Result section p-values were added

12. Terms such as “weak statistically significant”, “low statistically significant” or “little statistically significant” should not be used to describe the obtained results of experiments. Authors should use p-values in the description of the outcomes.

Terms such as “weak statistically significant”, “low statistically significant” or “little statistically significant” are no more used and p-values were added

13. Figures: It is not necessary to write the dose of the tested substance in each column, especially when it is the same everywhere. This information can be included in the caption to the figure.

The dose used is included only in the caption of the figure

14. The cited literature is quite old. Is there no more recent literature data?

We replaced the old literature with newer one

15. L. 29 “good” isn’t appropriate word if we are talking about the neuroprotective and antioxidant effects

We removed word good and are talking about the neuroprotective and antioxidant effects

16. L. 46 “treatment” isn’t appropriate word if we are writing about model, maybe it should be …include models induced by …

We replaced the word “treatment” with “The most commonly used models for PD include:…”

 17. L. 46 Please provide whether indicated models are animal or in vitro PD models?

Nowadays, there are a lots of in vitro and in vivo models which are used to mimic the pathological mechanisms in different neurodegenerative diseases. The most commonly used models for PD include: induction with reserpine (in vivo), 6-hydroxydopamine (6-OHDA) - in vitro and in vivo,  or 1-methyl-4-phenyl-1,2,3,6-tetrahydropyridine (MPTP) (in vitro and in vivo).

18. L. 49,52,56 There's something wrong with the citation

The reference style was corrected

Reviewer 4 Report (New Reviewer)

Oxidative stress is a phenomenon caused by an imbalance between production and accumulation of oxygen reactive species (ROS) in cells and tissues and the ability of a biological system to detoxify these reactive products. ROS can play several physiological roles (i.e., cell signaling), and they are normally generated as by-products of oxygen metabolism; despite this, environmental stressors and xenobiotics contribute to greatly increase ROS production, therefore causing the imbalance that leads to cell and tissue damage (oxidative stress). Oxidative stress is suspected to be important in neurodegenerative diseases including Lou Gehrig's disease, Parkinson's disease, Alzheimer's disease, Huntington's disease, depression, and multiple sclerosis. Therefore, in recent years several antioxidants have been exploited for their actual or supposed beneficial effect against oxidative stress. However, before publishing this manuscript, the Authors should pay attention to a number of important issues that prevent its publication.

In the abstract, the Authors write that creatine and its derivatives cause weak neurotoxic and pro-oxidant effects, however, these effects are not weak, but rather very significant.

The manuscript lacks information about used creatine derivatives - the method of synthesis, preparing solutions, etc. In addition, information on how they differ from each other and what significance these differences in structure have for their operation should be provided.

Line 168 – The Authors claimed that "substances maintained synaptosomal viability statistically significantly by 701% and 73% and GSH levels by 70% and 80%......" however, the graph shows statistical significance p<0.05. I suggest checking this fact again because with such high differences, the significance should be higher.

Interesting is the fact that on the figures the Authors presented data for various creatine derivatives and they are practically the same. How is it possible that there are almost no differences between them? How to explain it, what does it mean? I miss that explanation.

Line 233: the Authors claimed that creatine and its derivatives showed little statistically significant neurotoxic effect but previously explained that it is at the level of 20-30%. Therefore, in my opinion it is not a little. Please provide a comment on this.

The manuscript describes only the action of 6-OHDA, no characteristics of Fe/AA and t-BuOOH. Please complete this information.

Lines 233 to 244: The Authors emphasize the neurotoxic nature of creatine and its derivatives, so why did they decide to use them in research?

How the Authors will explain the pro-oxidative and damaging effects of creatine when administered to animals from the control group? How beneficial it is in animals from the study group? What is the mechanism of these differences? Can toxic effects be related to using to high dose?

The Authors should supplement the information on how they selected the dose - provide appropriate references and definitely expand the information on the tested derivatives. Did they have toxic effects at doses other than those used or were they already tested in other models?

Author Response

Journal name: Life

MS ID: life-2089599

Title: Evaluation of in vitro neuroprotective and antioxidant activity of newly synthetized creatine derivatives on different sub-cellular fractions (rat brain synaptosomes, mitochondria and microsomes)

Authors: Ivanka Kostadinova1, Magdalena Kondeva-Burdina1, Lyubomir Marinov1, Lyubomir L. Vezenkov², Rumyana Simeonova1*

We would like to thank the reviewer for reviewing our manuscript and providing constructive comments. The answers to the comments is detailed below.

We have added one more co-author - Lyubomir L. Vezenkov²

All corrections are colored in blue

Questions/ Comments

Answer

In the abstract, the Authors write that creatine and its derivatives cause weak neurotoxic and pro-oxidant effects, however, these effects are not weak, but rather very significant.

The term “weak” was removed

All corrections are colored in blue

The manuscript lacks information about used creatine derivatives - the method of synthesis, preparing solutions, etc. In addition, information on how they differ from each other and what significance these differences in structure have for their operation should be provided.

In the article, we provided a general formula of the newly synthesized creatine derivatives and data about the patent

Line 168 – The Authors claimed that "substances maintained synaptosomal viability statistically significantly by 701% and 73% and GSH levels by 70% and 80%......" however, the graph shows statistical significance p<0.05. I suggest checking this fact again because with such high differences, the significance should be higher.

After repeating the statistical analysis, we corrected the statistical significance and appreciate the reviewer's remark

Interesting is the fact that on the figures the Authors presented data for various creatine derivatives and they are practically the same. How is it possible that there are almost no differences between them? How to explain it, what does it mean? I miss that explanation.

Newly synthesized creatinine derivatives are structurally similar and this may explain the close results in terms of their toxicity

Line 233: the Authors claimed that creatine and its derivatives showed little statistically significant neurotoxic effect but previously explained that it is at the level of 20-30%. Therefore, in my opinion it is not a little. Please provide a comment on this.

We have changed the sentence into: “In our models, administered alone, creatine derivatives and creatine show statistically significant neurotoxic effect.”

The manuscript describes only the action of 6-OHDA, no characteristics of Fe/AA and t-BuOOH. Please complete this information.

We have added the following additional information: “Tert-Butyl hydroperoxide (t-BuOOH) is a classical lysosomal membrane permeabilization inducer that causes lysosomal damage via oxidative stress. Microsomes can be used as a lipid membrane model to evaluate potential antioxidant activity. When ferrous sul-fate is combined with ascorbic acid, a Fenton reaction takes place, which is associated with the generation of free hydroxyl radicals (•OH).”

Lines 233 to 244: The Authors emphasize the neurotoxic nature of creatine and its derivatives, so why did they decide to use them in research?

In the current study, we established these toxic effects of newly synthesized creatine derivatives. In the presented in vitro models of induced oxidative stress and neurotoxicity, these compounds show neuroprotective effects. Many of the drugs that are used in practice have shown significant toxic effects, e.g., when determining their acute toxicity in animals, but are used as agents for the treatment of some neurodegenerative diseases. A classic example is Galantamine, which is one of the approved drugs for the treatment of Alzheimer's disease.

How the Authors will explain the pro-oxidative and damaging effects of creatine when administered to animals from the control group? How beneficial it is in animals from the study group? What is the mechanism of these differences? Can toxic effects be related to using to high dose?

In our previous animal studies, these creatine derivatives did not show toxic effects, but in our current in vitro study, these toxic effects are most likely due to the concentration used.

The Authors should supplement the information on how they selected the dose - provide appropriate references and definitely expand the information on the tested derivatives. Did they have toxic effects at doses other than those used or were they already tested in other models?

The concentration used (38 μM) was calculated based on data from previous in vivo experiments. We didn’t investigate these compounds in other doses and there is not always a correlation between in vitro and in vivo toxicity.

Best Regards, 

Simeonova

Round 2

Reviewer 3 Report (New Reviewer)

The Authors considered all my comments and revised the manuscript accordingly.

Reviewer 4 Report (New Reviewer)

In my opinion this manuscript can be published in current version.

This manuscript is a resubmission of an earlier submission. The following is a list of the peer review reports and author responses from that submission.

Round 1

Reviewer 1 Report

Comments for Life-2016329

The manuscript aimed to in vitro evaluation of newly synthetized creatine derivatives for possible antioxidant and neuroprotective activity on isolated rat brain sub-cellular fractions – synaptosomes, mitochondria and microsomes. The content of the manuscript is interesting, but there are some problem in experimental design should be solved before the manuscript been considered for publication.

Substantial revisions

Q1: Although this study is based on animal protection regulations and has been reviewed by IACUC, please provide the IACUC number and the number of animals used.

Q2: Please add a discussion on the role of creatine derivatives in Ca2+ influx, zeta potential or firing activity (patch).

Q3: Please add a discussion on the relationship between amino acid ( lysinate, ornitate, leucinate, valinate) and brain mitochondrial activity, so that readers can better understand its importance.

Q4: Please provide a flow chart of the mechanism of action of antioxidative and neuroprotective in vitro about how rat brain synaptosomes, mitochondria, and microsomes were processed by 6-hydroxydopamine 60 (6-OHDA).

Q5: There is no graph in this study to show that creatine derivatives do not affect the morphology of synaptosomes, please add a description in the text.

Q6: All the pictures in the manuscript are not clear, please provide clear pictures and enlarge the font.

Reviewer 2 Report

Title:

1.     I think the authors need to remove the following words from the title “rat brain synaptosomes, mitochondria, and microsomes).

2.     Writing a summary is not a requirement of “Life”.

Abstract:

1. In Line 34, the authors have not provided the full form of the abbreviation “ROS”

2.     Authors should provide any implication of their study in the last lines of the abstract.

Introduction

1.     Is there a specific reason, the authors only mentioned Parkinson’s disease in the starting lines of the introduction? There are several neural disorders associated with oxidative stress. It seems that authors are studying some aspect of PD rather than the oxidative potential of creatine. Please explain this.

2.     What types of in vitro models are available to mimic the pathological mechanisms and to understand the neurodegenerative processes? At least a few should be mentioned. (Refer to lines numbers 58-59)

3.     Authors need to modify these sentences, as they look like a research report rather than a scientific research article (lines 65-72)

4.     Authors must discuss the importance of creatine and its derivatives and then its relationship with different biological mechanisms. However, they are first providing the secondary information and at the last of the introduction write about the actual topic of interest. The introduction needs to be rewritten.

Materials and methods:

1.     Sections 2.2 and 2.3 do not look like a part of a research article. Please rewrite these.

2.     At what temperature the pellet was frozen? (Line 171)

3.     What positive or negative control was used by the authors for determining GSH levels? (Section 2.5)

4.     Please correct the numerical values and properly write the scientific notation. (Line 174-175)

Results

1.     The text font in Figure 1, 2, and 3 is too small and hard to read. Please increase the font size and use a high-resolution image.

2.     Please explain lines 185-787).

3.     It would be much better if the authors provide the results in a tabular form for better comparison between the results of different derivatives.

Discussion

1.     In the first paragraph of the discussion authors are giving all the examples of athletes. Is there no prior study available with the rat model?

2.     Correct the name of hydrogen peroxide (line 267)

3.     Authors need to discuss the limitations of their study in the last paragraph of the discussion.

4.     Authors provided unnecessary detail about t-BuOOH (lines 308-330) and has written it like a research report rather than a scientific article.

Conclusion

1.     Authors should provide the implications of their study.

Reviewer 3 Report

Dr. Kostadinova et al., analyzed the in vitro neuroprotective and antioxidant effects of freshly synthesized creatine derivatives on various sub-cellular fractions (rat brain synaptosomes, mitochondria, and microsomes). It is reported that when administered alone on isolated rat brain subcellular fractions, creatine derivatives and creatine demonstrated modest neurotoxic and pro-oxidant effects. The derivatives demonstrated good neuroprotective and antioxidant properties in oxidative stress models of iron/ascorbate (on microsomes), tert-butyl hydroperoxide (on mitochondria), and 6-hydroxydopamine (on synaptosomes). Specifically, it is suggested by the Authors that these effects could result from reactive oxygen species scavenging and retention of the low glutathione level.

Major points.

1. The title of the manuscript and the introduction conflict with one another. The latter concentrates on Parkinson's disease. It is surprising, nevertheless, that the pathology is not stated in the title.

2.     At least in my word version, the figure quality is poor. I find it challenging to read the legends.

3.   It is not well explained how the compound investigated alone may be neurotoxic but neuroprotective when administered in combination with e.g. 6-OHDA.

4.  The authors' discussion of the effects of creatine derivatives and creatine on athletes in the Discussion while addressing PD in the introduction is a little confusing.

5. The bibliography contains obsolete and insufficient references.
